Integrating adaptation pathways and Ostrom’s framework for sustainable governance of social-ecological systems in a changing world

http://orcid.org/0000-0002-9276-8628 Pichancourt Jean-Baptiste 1 jean-baptiste.pichancourt@inrae.fr
http://orcid.org/0000-0002-6163-2824 Brias Antoine 2
Bonis Anne 2
1 Université Clermont Auvergne (UCA), INRAE, UR 1465 LISC , Clermont-Ferrand , France
2 Université Clermont Auvergne (UCA), CNRS, UMR 6042 GEOLAB , Clermont-Ferrand , France
Banaszak Anastazia
Electronic publication date: 2025 Feb 24
Publication date: 2025
Volume: 13
Electronic Location ID: e18938
Received 2024 May 17; Accepted 2025 Jan 15
Copyright: © 2025 Pichancourt et al.
Copyright year: 2025
Copyright holder: Pichancourt et al.
License: This is an open access article distributed under the terms of the Creative Commons Attribution License, which permits unrestricted use, distribution, reproduction and adaptation in any medium and for any purpose provided that it is properly attributed. For attribution, the original author(s), title, publication source (PeerJ) and either DOI or URL of the article must be cited.
License URL: https://creativecommons.org/licenses/by/4.0/

Keywords: Agroecosystem, Hedgerows, Biodiversity, Ecosystem services, SES analysis, Robustness framework, Complex dynamical system, Viability theory, DAPP maps

Funding: PACSEN Project European Regional Development Fund (European Union) FEDER Auvergne-Rhône-Alpes The work, including the master’s student and the post-doctoral fellowship of Antoine Brias, was supported by the PACSEN project (led by Anne Bonis) and funded by the European Regional Development Fund (European Union) and the FEDER Auvergne-Rhône-Alpes. The funders had no role in study design, data collection and analysis, decision to publish, or preparation of the manuscript.

==============================
Dynamic Adaptive Policy Pathway (DAPP) maps are used to plan management decisions in contexts of high uncertainty, such as those driven by environmental changes affecting critical assets. Recent discussions emphasize their relevance for addressing complex common-pool resource challenges, where diverse species, actors, and ecosystem services are intricately connected. However, designing DAPPs for such multifaceted social-ecological systems (SES) is challenging due to the extensive range of potential adaptation options. This study presents a general method to address these challenges by leveraging Ostrom’s theoretical frameworks for the governance of common pool resources—the Institutional Analysis & Development framework (IADF), the social-ecological systems framework (SESF), and the coupled infrastructure systems framework (CISF). These frameworks were used to design nested DAPP maps that structure a large number of adaptation actions across multiple levels of institutional arrangement (operational, collective-choice, constitutional), and then develop a mathematical model to analyze the dynamic robustness of a SES across all potential pathways. The method was applied to predict and understand DAPP maps for supporting the collective management of hedgerow networks delivering diverse ecosystem services. DAPP maps for two SES were compared—one rural and one peri-urban—in France’s agro-ecological landscapes of the Auvergne region. We further modeled the impact of climate change on hedgerows characterized by different size and species richness, revealing the sensitivity of these DAPP maps to transit between nine nested institutional arrangements. We discuss the methodological and practical implications of this approach for managing SES characterized by greater diversities of interconnected species, actors, and ecosystem services, highlighting its strengths and challenges in guiding adaptation under deep uncertainty.

Introduction

The Adaptation Pathways Framework (Wise et al., 2014; Werners et al., 2021) supports the collective exploration of adaptation options for vulnerable assets under global environmental changes, accounting for how time influences perceptions of future outcomes. This framework emphasizes the role of human imagination in envisioning incremental adaptation pathways and recognizes how consensual or conflicting expectations actively shape the complex realities of social-ecological systems (SES).

The Dynamic Adaptive Policy Pathway (DAPP) approach maps adaptation as sequences of potential actions, reassessed over time (Haasnoot et al., 2013; Haasnoot, Warren & Kwakkel, 2019). While DAPP maps are widely recognized for their flexibility and efficiency in diverse contexts (Werners et al., 2021), recent studies highlight limitations in addressing more complex SES (Hermans et al., 2017; Roelich & Giesekam, 2019; Stanton & Roelich, 2021).

Complex SES encompass a wide diversity of interlinked species, ecosystem services, actors, infrastructures, and institutional rules, all of which can shape the set of possible actions within DAPPs. Sustaining ecosystem service (ES) needs in such complex SES often requires simultaneous or sequential adaptation interventions across various ecological and social components (Allen et al., 2012). This complexity can result in an overwhelming number of possible pathways. For instance, a simple scenario involving the choice between 10 actions (targeting different species, ecosystems, ecosystem services and/or actors) reassessed over 10 time steps, generates 1010 (10 billions) pathways, each requiring viability testing.

To address the complexity of socio-ecological systems (SES), a series of qualitative studies conducted in the same agro-ecological region of the French Alps have developed simplified guidelines for designing DAPP maps (Colloff et al., 2016; Lavorel et al., 2019, 2020; Bergeret & Lavorel, 2022). Although these studies were not explicitly framed using Ostrom’s governance theory of common-pool resource (CPR) (Ostrom, 1990), their findings collectively align remarkably well with Ostrom’s principles. Specifically, they illustrate how authority can be progressively devolved, as collectives transition from formulating broad adaptive visions (Colloff et al., 2016; Lavorel et al., 2019) to implementing operational actions for on-ground CPR management (Lavorel et al., 2020; Bergeret & Lavorel, 2022). However, these studies stop short of explicitly addressing how devolution and horizontal interactions should be adapted within and between hierarchical levels of institutional organization to facilitate polycentric governance in adaptation efforts. Ostrom’s theory emphasizes the importance of nested governance, identifying four interrelated levels that could serve as a structure for adaptation decisions and rules in complex SES. By integrating these nested levels, the above-mentioned works could have more effectively demonstrated how to structure the processes and interactions that underpin sustainable governance and adaptation in polycentric systems, notably:

1. Level of meta-constitutional-choice arrangements (MCA): adaptation begins with broad shifts in societal visions or worlviews, each prioritizing different values and objectives (e.g., Colloff et al., 2016; Lavorel et al., 2019, 2020; Bruley et al., 2021; Bergeret & Lavorel, 2022). For example, rural communities adapting to urbanization or mountain regions adapting to climate change may require a redefinition of institutional priorities, such as in functional bio-diversity types, ES needs, landscape archetypes, or broad definitions of institutions and rule clusters (Lapointe, Cumming & Gurney, 2019; Lavorel et al., 2019; Bergeret & Lavorel, 2022).

2. Level of constitutional-choice arrangements (CCA): Within each MCA, multiple CCA can be envisioned, redistributing authority, rights and roles among actors and infrastructures. These arrangements determine who is eligible to manage ES and in what capacity (cf. Ostrom, 1990). For example, Bergeret & Lavorel (2022) show that different actor combinations may prove more effective depending on the prevailing MCA vision.

3. Adaptation of collective-choice arrangements (KCA): within each CCA, different networks of eligible actors, infrastructures and roles can be envisioned to manage ES resources, resulting in different KCA. However, “reaching a consensus” on new KCA is often a significant challenge for SES adaptation (Bruley et al., 2021), especially when roles and responsibilities remain unclear.

4. Adaptation of operational-choice arrangements (OCA): The final level, within each KCA, involves adapting the target, frequency or intensity of specific operational actions, such as species harvesting, monitoring, or regulating resource access. For example, Bergeret & Lavorel (2022) identified 327 possible adaptation of operational actions (among three MCA), which would involve altering ecosystem variables directly, or enabling self-transformation through interventions like creating semi-natural infrastructures (like grasslands or hedgerows) that are species-rich and functionally connected (e.g., Lavorel et al., 2020; Bruley et al., 2021).

Given the structure of DAPP maps, it becomes evident that organizing adaptation actions across multiple levels of governance (OCA, KCA, CCA, MCA, etc.) in a way that accounts for unforeseen feedback and spillover effects is challenging. Additionally, comparing DAPP maps across case studies is impractical without a standardized approach. Navigating this hierarchy of nested governance structures is crucial for understanding and securing effective adaptation. Actions at lower levels may compromise long-term sustainability at higher levels, and vice versa, highlighting the need for alignment across governance scales (Ostrom, 1990). Standardizing the design of nested DAPP maps would improve their ability to capture complex systemic interactions, better explain emerging adaptation challenges, and facilitate meaningful comparisons across case studies.

Theoretical principles and analytical frameworks developed by Ostrom may provide a robust foundation for designing such nested DAPP maps. The Institutional Analysis and Development Framework (IADF) (Ostrom, 1990) examines how clusters of institutional rules and sequences of adaptive policy decisions shape actor behavior within action arenas, drive SES outcomes, and create feedback loops at different level of institutional organization. In theory, this temporal and iterative structure aligns well with the DAPP approach. Building on the IADF, two derivative frameworks may also enhance its applicability to DAPP maps. The social-ecological systems framework (SESF) (Ostrom, 2007) was designed to refine the IADF by identifying key variables—such as resource units, governance systems, actors, and infrastructures—that influence SES adaptive capacities or can be targeted for adaptation. The coupled infrastructure systems framework (CISF) (Anderies, Janssen & Ostrom, 2004) extends these insights by focusing on the interactions between social and ecological infrastructures and human actors within nested institutional arrangements. By representing polycentric governance models as systems of equations (Muneepeerakul & Anderies, 2020), these frameworks allow for the prediction of governance capacities to prevent failures and maintain long-term system robustness under external stressors.

Objective and structure of the article

Building on these theoretical foundations, we aim to integrate Ostrom’s frameworks with mathematical methods to enhance the design of nested DAPP maps and deepen the understanding of complex SES adaptations across multiple levels of organization. The methodology section elaborates on the general formalism, detailing the steps and mathematical links between rule clustering at different levels for the IADF, SESF, CISF, and DAPP.

We apply this method to an agro-ecological landscape characterized by species-rich hedgerow networks that provide a bundle of ES to the local community. Our case study focuses on two archetypal SES—peri-urban and rural—each defined by distinct MCA visions of ES needs. We also account for variations in hedgerow height and plant species richness, which are differentially influenced by human management and climate change, and in turn, affect the dynamic delivery of ES in our model. The resulting DAPP maps highlight the adaptive pathways required to address the challenges of climate change, particularly drought stress, on hedgerow functionality. These maps reveal nested levels of viable adaptation and capture differences in SES archetypes under varying climate stress scenarios.

Finally, we identify key indicators that elucidate the primary drivers influencing DAPP map variations and adaptation outcomes, offering insights into the complex interplay between nested human management, ecosystem functionality, and climate change.

Methodology

We used a six-step workflow to show how to mathematically derive such nested DAPP maps from Ostrom’s theoretical frameworks. The process begins by applying the SESF to redefine the possible set of adaptation actions used in DAPP, based on targeted attributes within the SES. Next, the CISF serves as a secondary filter to redistribute adaptive action targets according to the roles attributed to various actors and infrastructures, across nested levels of institutional arrangements (OCA, KCA, CCA, MCA, etc.). This step aligns with Ostrom’s principles of polycentric and nested governance (Ostrom, 1990). Subsequent steps transform the nested CISF representations into equations describing SES dynamics (Muneepeerakul & Anderies, 2020), link these equations to viable control theory (Aubin, Bayen & Saint-Pierre, 2011), and infer a mathematical definition and graph-theoretical representation of viable or optimal nested DAPP maps. Finally, we demonstrate how to retrospectively analyze the role of specific actions in shaping pathway viability.

Step 1. Using the SESF for defining the set of actions based on adaptation targets

The first step consists in defining the set of n possible adaptation actions U = {u1, u2, …, un}, according to the attributes of the SES. The importance of certain attributes can change with the SES context. Selecting the appropriate target, trigger and scale of these actions necessitates a comprehensive description of the SES attributes, which was made possible with the use of Ostrom’s SESF (Ostrom, 2007, 2009).

The SESF—derived from the simplification of the 450 attributes presented in Ostrom’s CPR codebook (Ostrom et al., 1989)—provides a reduced set of 1st tier SES attributes, that can be used as potential action targets. This set can be further unpacked into 2nd, 3rd, 4th, etc… tier attributes, if the context and data availability require it. Different iterations of the SESF multi-tier structure were created (Ostrom & Cox, 2010; Basurto, Gelcich & Ostrom, 2013; McGinnis & Ostrom, 2014; Vogt et al., 2015), but we were interested in one that could analyze SES rich in species, ES and actors.

Step 1.1. Definition of the 1st tier action set

For the 1st tier, we used the SESF version developed by McGinnis & Ostrom (2014), emphasizing the generalization from users to actors attributes. As such, we defined U as a set of eight possible sub-sets of actions targeting different elements of the SES, such as U = {US, UGS, URS, URU, UGS, UA, UI, UO} (see Table 1 for definitions of subsets).

Table 1 Definition of the 2nd tier action set U based on the social-ecological system framework (SESF).

Set U of 2nd tier actions based on a version of Ostrom’s social-ecological system framework (SESF)	
Set of actions related to the social, economic & political settings (U S )	
US1	Economic development*,†	US4	Other governance systems†	
US2	Demographic trends*,†	US5	Markets†	
US3	Political stability (rate of political change)*,†	US6	Media organizations†	
		US7	Technology†	
Set of actions related to the resource systems (U RS )	Set of actions related to the resource units (U RU )	
URS1	Sector	URU1	Resource unit mobility*,**,†	
URS2	Clarity of system boundaries	URU2	Growth or replacement rate of resource units	
URS3	Size of resource system	URU3	Interactions among resource units*,†	
URS4	Human constructed facilities*	URU4	Economic value	
URS5	Productivity of the system	URU5	Number of units	
URS6	Equilibrium properties	URU6	Distinctive characteristics	
URS7	Predictability of system dynamics	URU7	Spatial & temporal distribution	
URS8	Storage characteristics			
URS9	Location			
URS10	Ecosystem history			
Set of actions related to the governance systems (U GS )	Set of actions related to the actors (U A )	
UGS1	Policy area	UA1	Number of relevant actors*	
UGS2	Geographic scale of governance system*	UA2	Socio-economic attributes*	
UGS3	Proportion of participating population**	UA3	History of past experience*,**	
UGS4	Regime type (demo/auto-cratic, mono/poly-centric)*	UA4	Location*	
UGS5	Rule-making organizations*	UA5	Leadership/entrepreneurship*	
UGS6	Rules-in-use*	UA6	Norms/(trust-reciprocity)/social capital*	
UGS7	Property rights systems (relations among people in relation to resource units and infrastructures)*	UA7	Knowledge of SES/mental models/beliefs*	
UGS8	Repertoire of cultural knowledge, beliefs, norms, practices (strategies) with no rules and sanctions*	UA8	Proportion of resource dependent actors*	
UGS9	Network structure (connections among the rule-making organizations and the population subject to these rules)*,**	UA9	Technologies available*	
UGS10	Historical continuity of the governance system (recent vs long-lasting, open vs close to internal adaptation)*			
Set of actions related to the interactions (UI)	Set of actions related to the outcomes (UO)	
UI1	Harvesting/using resource units by divers users	UO1	Social performance measures (e.g., efficiency, equity, accountability, sustainability)*,**	
UI2	Information sharing among actors*,**	UO2	Ecological performance measures (e.g., overharvested, resilience, robustness, biodiversity)**	
UI3	Deliberation process*,**	UO3	Externalities to other SES†	
UI4	Conflicts among actors*,**			
UI5	Investment activities*,**			
UI6	Lobbying activities*,**,†			
UI7	Self-organizing activities*,**			
UI8	Networking activities*,**			
UI9	Monitoring activities*,**			
UI10	Evaluative activities*,**			
Notes:

* Indicates that the action set is decomposed into three 3rd tier action subsets targeting actors and infrastructures with exploitation (E), conservation (C) and policy-making (P) roles.

** Signifies that the 3rd tier set is extended to account for the directional influences of one 3rd tier attribute on the other, or also the resource system (RS).

† Indicates the decomposition of 3rd tier into 4th tier action subset to account for the level of institutional arrangement that trigger this action: operational-choice arrangement (OCA), collective-choice arrangement (KCA), constitutional-choice arrangements (CCA), meta-constitutional arrangement (MCA), etc.

Step 1.2. Definition of the 2nd tier action set

The 1st tier action set was then decomposed to emphasize a 2nd tier of possible action sets. For instance US∈U was defined as US = {US2, US3, US4, US5, US6, US7, US8, US9} as per Table 1, emphasizing the possible actions involving various external elements impacting/ed on/by the SES. Some second tier attributes were modified using Vogt et al.’s (2015) SESF version, in order to better integrate ecological and ES attributes as possible targets of adaptation (see Table 1). We particularly transformed its 2nd tier attribute “externalities to other SES” (O3) into a 3rd tier action (see step 1.3) to emphasize the role of ecological resource infrastructures and associated species.

Step 1.3. Definition of the 3rd tier action set

We further introduced a 3rd tier action subset in anticipation of the integration of the action sets into the CISF (step 2). For instance, we subdivided certain 2nd tier attributes (see action sets in Table 1: US1:US3, URS4, URU1, URU3, UGS2:UGS10, UA1:UA9, UI2:UI10, UO1) into three new 3rd tier attributes to emphasize the fact that actions can target different roles attributed to actors (A in Table 1), to infrastructures from the resource and governance systems (RS, GS), interactions (I) and outcomes (O). The 3rd tier action set U emphasized the three following roles:

(i) Resource exploitation role (E) attributed to exploiting actors ( UE:A2), social infrastructures (e.g., norms and social capital, UE:A6), of institutional infrastructures (e.g., rules-in use, strategies: UE:GS1 to UE:GS10) and of physical infrastructures (such as supply chains or exploiting technologies, UE:S7). These attributes involve various social-ecological interactions ( UE:I1 to UE:I10) associated with the hedgerows’ resource appropriation, provisioning, production, distribution/supply (chain), transformation, consumption/use and monitoring of own cost-effectiveness. They also involve various resource outcomes ( UE:O1 → UE:O3) in relation with the production of the desired ES. These attributes can all be related with the sustainable dimension of the resource utilization, provided they solely benefit from utilitarian objectives.

(ii) Resource Conservation role (C) (3rd tier action set noted e.g., UC:A1) is attributed to actors and infrastructures involved in supporting, maintaining, and monitoring resource systems to uphold broader societal and ecological values. The C-role often encompasses resource management driven by intrinsic ethical or aesthetic motivations—such as valuing life for its own sake, preserving cultural heritage, or safeguarding landscape beauty. Additionally, the C-role extends to the management of ES with utilitarian benefits that go beyond the immediate resource owners. These benefits may serve local communities, national interests, or even the global population. Examples include actions to prevent wildfires and landslides that could threaten nearby villages or enhancing carbon sequestration to mitigate global climate change.

(iii) Policy-making role (P) (3rd tier action set noted e.g., UP:A2) refers first to the role of support for E and/or C roles. It also refers to the role of arbitration and conflict-resolution between E and C roles, when trade-offs emerge between purely utilitarian E values and intrinsic or collective C values. Thirdly, it refers to the role of triggering adaptation actions during the DAPP process, like when changing operational, collective or constitutional arrangements.

It is important to note that SES do not necessarily possess E, C, and/or P role attributes and associated actions. For instance a resource free of humans do not have E, C and P. Furthermore, all the combinations between the three can in theory be found.

Following the same logic of defining these three roles targeted by these actions, we further specified the role that triggers these actions on the targeted role. For instance, the C-role can trigger an action targeting the economic attributes (A2) of the E-role. Using the SESF logic, this action was encoded as UC→E:A2.

Step 1.4. Definition of the 4th tier action set

We then defined a 4th tier action set, relative to the nested levels of adaptation initiatives operating at different time scales.

Using examples from Ostrom (1990) that were presented in the introduction, we define the adaptation actions at the level of the operational-choice arrangement (UOCA), of collective-choice arrangement (UKCA), of constitutional-choice arrangement (UCCA), meta-constitutional-choice arrangement (UMCA), etc. These levels are nested and can be attributed to any kind of roles defined in step 1.3 (cf. Table 1 for the sub-set possessing this 4th tier attribution). For instance, UC→E:A2OCA1→OCA2 means that the adaptation of OCA1 into OCA2 consists in changing the action that role C triggers on attributes A2 of E-role. Using the same logic, we will see in step 2.2 how to define adaptation actions at other upper levels of institutional arrangements (KCA, CCA, MCA, etc.).

Step 2. Using the CISF for a systemic organization of actions between roles

Step 2 consists in using the systemic representation provided by the CISF to organize the actions defined in step 1 between roles. The CISF, as originally defined by Anderies, Janssen & Ostrom (2004), represents any SES as an interacting system of potentially four compartments: One ecological resource compartment (R), and three social role compartments, originally referred to as the resource users (RU), the public infrastructures (PI) and the public infrastructure providers (PIP). These three compartments represent three role decision centers that can compete and/or cooperate, reflecting the polycentric nature of the action arena of the IADF around the resource (Anderies, Barreteau & Brady, 2019; Muneepeerakul & Anderies, 2020). Their interactions, represent actions that are encoded using specific numbers (cf. Anderies, Barreteau & Brady, 2019, and Fig. 1), which can be linked to the encoding of the SESF and CPR frameworks developed by Ostrom, as outlined by Anderies & Janssen (2013) and Bernstein et al. (2019).

Figure 1 Systemic modeling of adaptation actions using the coupled infrastructure system framework (CISF).

Compartments represent ecological and social roles: the resource role (R) includes species, ecological infrastructures (e.g., forests, hedgerows, grasslands, rivers), and associated ecological processes, while the exploitation (E), conservation (C), and policy-making (P) roles involve actors, infrastructures, and actions. Internal dynamics of each role’s capacity for action are shown as self-loops (i.e., action sets U0a→U0d). Interactions between compartments are represented by action sets (U1a→U1b to U6a→U6b), and with the broader environmental settings by U7a→U7g. References to action sets from the SESF (see Table 1) are included in brackets, e.g., U2a corresponds to UP→E from the SESF (see step 1.3) and can be associated with targeting many attributes of the B-role (e.g.,). An asterisk (*) indicates that the action set is decomposed into three 3rd tier action subsets targeting actors and infrastructures with exploitation (E), conservation (C) and policy-making (P) roles. Two asterisks (**) signify that the 3rd tier set is extended to account for the directional influences of one 3rd tier attribute on the other, or also the resource system (RS). A dagger (†) indicates the decomposition of 3rd tier into 4th tier action subset to account for the level of institutional arrangement that trigger this action: operational-choice arrangement (OCA), collective-choice arrangement (KCA), constitutional-choice arrangements (CCA), meta-constitutional arrangement (MCA), etc.

Step 2.1. Defining the model used for the CISF

The CISF is flexible and different model implementations can be defined. Here we defined the three interacting compartments RU, PI and PIP has representing the three interacting roles that actors and governance systems’ infrastructures possess (respectively E, C and P), as defined in step 1.3. This way, each role compartment could possess its own model of representation (as defined in Aggarwal & Anderies, 2023), set of actors and infrastructures (like in Pichancourt, 2023, 2024). Furthermore like in Muneepeerakul & Anderies (2020) or Pichancourt (2023, 2024), the same actor and infrastructure can possess multiple roles (E, C and P), whose rate of implication or interaction rate is modeled using any of the following linking action sets U2a, U2b, U3a, U3b, U6a, U6b from Fig. 1. For example, actors and private/shared infrastructures can simultaneously be attributed to (i) a supply chain role in E (e.g., as farmers and intermediary supply chains from farm to fork, forest harvesters with their sawmills and tracks, fishermen from the fleet to the auction house and fish market), (ii) a conservation role in C (e.g., as neighboring group of farmers in partnership with public, private or NGOs conservation agencies), and (iii) a policy or arbitration role of P (e.g., as a board member of a governing body that is buying lands and renting them to farmers, and set specific rules and granting schemes). Using this encoding we attributed multiple actions sets from the SESF to one code of action defined in the CISF (Fig. 1).

Based on this structure (cf. Fig. 1), we then model how the different roles interact with the external context (i.e., associated with action set US from Table 1), using any of the set action links U7d, U7f, U7h. For instance, an external NGO can offer three distinct technological training schemes to increase the capacity of action of the three role compartments (i.e., US7 from Table 1): one for actors with exploiting roles on how to sustainably exploit their private resource R for private or club ES benefits ( U7d = UE:S7); another for actors with broader conservation role on how to collectively monitor or restore the state of R for broader public and common ES outcomes (U7f = UC:S7); and a third for actors or infrastructures with broader governing role to establishing DAPP rules at a certain level (U7h = UP:S7).

Step 2.2. Define adaptation actions per level of governance arrangement

In step 2.2 we use the CISF to define adaptation actions at different level of governance arrangement (i.e., OCA, KCA, CCA, MCA, etc.). Different CCAs of interest can be defined by specifying the eligibility criteria for certain roles within the CISF. This can be achieved by switching on and off specific roles as needed (see Fig. 2).

Figure 2 Adaptation at the level of constitutional-choice arrangements (CCA).

Define the set of CCA and adaptation actions between them, using four examples of CCA. The elements in red refer to the CCA level of adaptation, i.e., the type of constitutional-level actions that are required for the policy-making role P to trigger a constitutional change (e.g., from CCA:1 to others with action set UCCA:1→2,3,4 ), and influence or respond to the need of other roles E and C when implementing this change (e.g., action sets UCCA).

For every CCA, adaptation actions at a KCA level are modeled by switching on and of certain interactions between role compartments, leading to different coordination networks and chains of actions between actors and infrastructures, within and between roles (Fig. 3).

Figure 3 Adaptation at the level of collective-choice arrangements (KCA).

Define the set of KCA and adaptation actions between them, using three fictitious examples. These adaptation actions deal with changing the composition (chain, network) of actions between the illegible roles defined in step 2.2.1. Here, every KCA is associated with its mathematical expression for the composition of action (red color), and the set of possible actions from one KCA to the others (gray color).

Finally, for every KCA, different OCA and their transition can be modeled by changing the acceptable range for the intensity or frequency of operational actions (Fig. 4).

Figure 4 Adaptation at the level of operational-choice arrangements (OCA).

Define the set of OCA and possible adaptation actions between them, using two fictitious examples. These adaptation actions deal with changing the intensity, frequency or type of each operational action. Here, every OCA is associated with the set of possible actions from one OCA to another (gray color).

Step 3. Transformation of the CISF model into a set of equations

The interest of formulating a model based on the CISF, is that it systematizes the modeling of possible systemic adaptation and spillover problems (Anderies, 2015; Anderies, Barreteau & Brady, 2019; Houballah, Cordonnier & Mathias, 2020), and transforms them into a set of equations. By doing so we can predict infrastructural vulnerabilities related to ES production and external disturbances (Anderies, 2015; Muneepeerakul & Anderies, 2020). When combined with the viable control theory (Aubin, Bayen & Saint-Pierre, 2011; Martin, Deffuant & Calabrese, 2011), we can use the CISF to estimate system-wide metrics of robustness or resilience (Muneepeerakul & Anderies, 2017; Homayounfar et al., 2018; Houballah, Mathias & Cordonnier, 2021).

To create a system of equations, every action set at an operational level (OCA) must be defined as rates impacting the state of the targeted role. But every action and rate defined within the CISF (Fig. 1) can denote distinct interpretations through the SESF (Table 1), and thus algebraic formulations. Consequently, modelers need to make deliberate choices regarding their specific definition, like we did in the application example (see Supplemental Material). Nevertheless, some general recommendations can still be formulated in agreement with the general context of the article.

First, R can be defined through its potential for producing ES, ranging from 0 to +1 for each of them. The ecological dynamics of R (i.e., the action of R on itself, noted U0a) and actions sets U1a, U4b and U7b impact the potential of R for producing ES, all ranging between [−1, +1] (see Fig. 2). Similarly, for each social role compartment ( B, C, and P), the actions directed toward them define the relative capacity of actions gained or lost, within the range [−1, +1] or [0, +1]. Self-loops ( U0b, U0c, U0d: [−1, +1]) denote the natural growth or decay of the capacity of action of the respective roles B, P and C. This could be negative (loss through natural death or socio-cultural or economic instability), positive (e.g., gain through internal creativity), or neutral based on precise definitions.

Furthermore, action set U1a represents the management rate [0, +1] by E on R, while U1b represents the action set describing the extraction, supply, and transformation of a unit of ES into useful benefit for the state of R [−1, +1]. Similarly, U2a represents the action that improves the P-state when monitoring the E-state [0, +1], whereas U3b represents the same type of action that increases or decreases C-state when interacting with P [−1, +1]. U4a represents the action that increases C-state through monitoring R-state [0, +1], and U4b denotes the action of C to support, increase or restrict the rate of change 0a of R [−1, +1]. Action set U6a reflects the C-state gains or loss when E interacts with C; whereas U6b can represent the action that leads E-state gain or loss through support or sanctions by C [−1, +1]. U5a and U5b denote E-state gains from monitoring action sets U1a and U1b, respectively, within the range of [0, +1]. Similarly, U5a’ and U5b’ represent the regulation rate on respectively U1a and U1b [0, +1]. U7b signify the action of the external settings of R on its dynamics, e.g., climate stressing (acting) R by affecting natural resource infrastructure (e.g., hedgerows and associated species) producing ES [−1, 0]. Additionally, U7d, U7f, and U7h indicate E, C and P state gains, respectively, resulting from external socio-economic or climate factors or actions. Finally, U7a, U7c, U7e, and U7g represent externalities (processes or actions) of ES flowing to other SES outside the one studied [−1, +1].

Altogether, the total weight of all linking actions (which includes self-loops) directed toward each of the four compartments, adds up to 1, such that: for E: U0b + U1b + U2b + U6b + U7d = 1, for P: U0c + U2a + U3b + U7f = 1, and for C: U0d + U3a + U4a + U6a + U7h = 1.

Based on these general definitions—and following the CISF methodology described by Anderies, Janssen & Schlager (2016), Muneepeerakul & Anderies (2017, 2020), and Houballah, Mathias & Cordonnier (2021), Houballah, Cordonnier & Mathias (2023)—we obtain the most general system of equations describing the processes link the R state dynamics to the state dynamics of E, C and P; taking into account possible uncertainties:

(1) {dRdt=U0a⋅R⏞NatrualESGrowth±U7b⋅R⏞ImpactofSESSettings−U7a⋅R⏞ESExternalities(1a)+U4a⋅R⋅C⏞RSupportedbyC−U1b⋅R⋅E⏞ESAccess&Flow⋅U5b⋅C⏞Regulation+U1a⋅E⋅R⏞Access&Management⋅U5a⋅C⏞RegulationdEdt=±U0b⋅E⏞NatrualGrowth/Decay±U7d⋅E⏞ImpactofSESSettings−U7c⋅E⏞Externalities(1b)+U1b⋅R⋅E⏞AccessR&ESFlow⋅U5b⋅C⏞Regulation+U6b′⋅E⋅C⏞CSupport&Regulation±U2b⋅E⋅P⏞PSupport&SanctionsdCdt=±U0d⋅C⏞NatrualGrowth/Decay±U7h⋅C⏞ImpactofSESSettings−U7g⋅C⏞Externalities(1c)+U1b⋅R⋅E⏞Access&ESFlow⋅U5b′⋅C⏞Monitoring+U1a⋅R⋅E⏞Access&Management⋅U5a′⋅C⏞Monitoring+(U6a−U6b)⋅E⋅C⏞EJoining/leaving±U3a⋅P⋅C⏞PSupport&SanctionsdPdt=±U0c⋅P⏞NatrualGrowth/Decay±U7f⋅C⏞ImpactofSESSettings−U7e⋅C⏞Externalities(1d)±U2a⋅E⋅P⏞Joint/Support/Sanction±U3b⋅C⋅P⏞Joint/Support/Sanction.

Step 4. Analysis of the viability of various CIS arrangements under different levels of global stressors

Step 4.1. Definition of the constraint domain of satisfaction for the levels of ES

Based on Eq. (1), we can evaluate the congruence between costs and benefits for all the possible scenarios of CIS arrangements (CCA, KCA and OCA), as specified by Ostrom’s second design principle of good governance (Ostrom, 1990). To achieve this, we must respect the basic condition that the state turnover of R, such that:

(2) dRdt≥0⇒U0a+(U1a.U5a′+U4b)C≥U7b+U1b.U5a.

From Eq. (2) and Fig. 2, we see that the potential resource turn-over rate (dR/dt) depends on both the intrinsic rate of increase of R (U0a), and the conservation actions originating from either E through U1a or from C through U4b.

Now imagine that the E-state is defined by the level of n ecosystem services ES1, ES2,…, ESn, exploited by E. The state space of R is now defined as SR={ES1,ES2,…,ESi,…,ESn}∈Rn. Equation (1a) would be thus dR(ES)dt=(dES1dt,dES2dt,…dESndt).

Population needs would be defined as viable iff the delivery of n ecosystem services (ES+) or disservices (ES−) respect the following constraints: ES+i ≥ ES+i,min and ES−j ≤ ES−j,max, where ES+i,min and ES−j,max represent the minimal and maximal acceptable value of ES i or j respectively associated with actors’ needs. Together for the n ES, these threshold form a constraint domain KR, that we refer to as the satisfactory domain of R, such that KR={(ES1,ES2,...,ESn)∨ESimin≤ESi≤ESimax∀i,K∈SR}.

Step 4.2. Definition of the set of viable trajectories of resource that respect KR

Based on the Eq. (2) and KR, we now define robustness (or lack thereof) as the subset of the state space for which there is at least one sequence of adaptive governance action u(.)∈A(T)fort∈[0,T], starting from the initial state, that robustly keeps the SES’s trajectory within KR, for any time step t during the time horizon T. As such, and following Aubin, Bayen & Saint-Pierre’s (2011) Viable Control Theory, we state that u(.) is viable (i.e., robust1 ) and the set of all viable u(.), is called the viability kernel ViabKR defined as:

(3) ViabKR(T)={R(t=0)∈KR:∃u(.),suchthat∀t∈[0,T],R(t)∈KR}.

In the most general case specified after Ostrom (1990), the nested sets of adaptive actions of governance changes are defined for u(t) ∈ UOCA ∈ UKCA ∈ UCCA ∈ UMCA… ∈ U, such that:

(4) u(.)=(uCCA:KCA:OCA(0),uCCA:KCA:OCA(1),...,uCCA:KCA:OCA(t),...,uCCA:KCA:OCA(T)).

In practice, we efficiently compute ViabKR for Eq. (1) representing the dynamics of the SES, using the Saint-Pierre backward algorithm on a discrete grid of the state space (Aubin, Bayen & Saint-Pierre, 2011).

Step 5. Deducing the DAPP map from the subset of viable solution

From Eqs. (3) and (4), we then define the regulation map RK (cf. Aubin, Bayen & Saint-Pierre, 2011: definition 2.9.4. p.73) used for mapping any state of R in KR to the subset RK consisting of controls U which are viable, in the sense that the corresponding direction dR/dt (cf. Eq. (1a)) is viable at any given point of the state space, such that:

(5) R(R(t))={u∈U(R):R∈KR,dRdt=f(R(t),u(t))∈ViabKR}.

We then transform Eq. (5) into a DAPP map, as represented by Haasnoot, Warren & Kwakkel (2019), by defining it as a directed graph G = (V, L), where: V→U×T are vertex/nodes at times t0,t1,…,tT for every possible action u(t), such that V=T∪t=0U(R(t))

L = {eij} ⊆ V × V are links (i.e. edges between vertices) representing possible viable state transitions between decision nodes under control u(t), such that: L=u(t),u(t+1)|u(t)∈RK(R(t),u(t)),u(t+1)∈RK(R(t+1),u(t+1))

The possible number of viable pathways represented in the graph will thus be equal or smaller than dim(G)max = UT.

Step 6. Assessing pathway sensitivity to social-ecological changes

Once we defined the viable DAPP map, we analyzed for every DAPP map, which actions explained retrospectively differences in viable outcomes, i.e., dViab(R)du(.). This step is detailed in Supplemental S2 in relation with the constraints associated with the application example.

Application example

Specification of the application example

The method was applied to a case study involving two SES sites in central France (one peri-urban and one rural), each with distinct visions and needs regarding ES, corresponding to different history and MCA. For each SES, adaptation actions were defined at the CCA and KCA levels, while OCA actions remained fixed for every KCA. Operational actions focused on managing a hedgerow network, with transitions between different hedgerow states defined by height and plant species richness (and the presence/absence of hedgerows). Since these states produced nine ES and were influenced by climate change, we compared DAPP maps across three climate stress levels for two SES.

DAPP maps were organized around nine potential scenarios of nested governance (CCA|KCA), with adaptations considered every five years over a 30-year period. This resulted in 4,782,969 (97) possible governance pathways, each evaluated for viability.

With this approach, we modeled DAPP maps where, for every step, we identifed the type of nested governance arrangement (CCA, KCA and OCA). We then determined what these adaptations assumed in term of changes in the SES structure (through the CISF). We finally checked the underpinning definitions of the SES attributes (using Ostrom’s SESF) that these adaptations implied.

We provide in supplementary files details on how we performed the SESF analysis (S1), gathered ecological data to model the resource dynamics (S2), applied the methodological steps to create DAPP maps (S3) and coded the model into Gnu-Octave (Eaton et al., 2024) to produce the results (S4). The main results are summarized below, with S5 providing supplementary results.

Five-year time-step evaluation of the viability and security of governance pathways

DAPP maps presented in Fig. 5 globally confirm our expectation that keeping the same governance arrangement for 30 years (especially CCA1, the one most frequently observed in our study site) is not predicted to be viable. They show on the contrary, that multiple adaptations are required, involving sequences of transitions between various combinations of CCA/KCA.

Figure 5 Probability DAPP maps.

Probability to find viable pathways of governance adaptation that meet the set of satisfaction constraints for all ecosystem services (ES), for two types of SES, three climate stress levels, and nine nested governance scenarios. Every panel provides the percentage of total viable trajectories starting from the state of the SES in 2020. A total of 97 = 4,782,969 pathways were estimated by combining CCA and KCA over the 30-years period and transitions every five years, i.e., between two successive decision nodes. Pathway sequences that are not represented in dark represent (irreversibly) non-viable five years sequences regarding the studied ES. Darker segments (on a 0–1 white and black scale) mean greater number of such unique 30-year long viable pathways crossing the 5-year segment between two decision nodes.

We decomposed the possible 30-year pathways into succession of 5-year pathway segments between two decision steps/nodes. We show that some of these segments are crossed by many unique 30-year pathway options, offering thus a broader range of future adaptation options. The darker segments on Fig. 5 seem less sensitive to an increase in climate stress level (0, 1, 2), but are sensitive to change in the SES context (peri-urban, rural).

The diversity of viable pathway segments and transitions options also changed with the SES type. More specifically, peri-urban SES (Figs. 5A–5C) offered a greater choice of viable pathways than rural SES, especially between viable KCA options within CCA2,3 or 4 options. This pattern was pretty insensitive to the increase in the level of climatic stress. This matched with the greater constrains on the satisfactory space that characterizes rural SES (see Table S2 in Supplemental S2). Accordingly, in rural SES, fewer satisfactory options of KCA transitions per CCA option were predicted (Figs. 5D–5F). There, actors would have to accept more drastic CCA transitions in order to respect the limits of the ES satisfactory domain. This is true in particular with no additional climate stress (Fig. 5D), as actors do have to first transit through CCA-B2. This first transition corresponds to contracting with the state government to become eligible for payments for ecosystem services (PES) (Fig. 4D, see links 2a,b in Fig. 2B). Then, 20 years later, we predict that viability maintenance of the rural SES requires to transform CCA2|KCA2 into CCA4|KCA8. This transformation is more demanding than from CCA1 to CCA2|KCA2, as it involves the setting of a new P-role for arbitration, collective rules and economic support between C and E (cf. Fig. 1). Unexpectedly, increased drought stress is predicted to diversify the number of KCA options, especially within the CCA3 option (Figs. 5E, 5F). However this larger choice is expected to come at the expense of the security level for every KCA choice (lighter gray shade), making those KCA transitions riskier.

The most secured 30-year viable decision pathways

We then selected the top 10% most secured options of viable 30-years adaptation pathways within the ES satisfactory domain out of the options in Fig. 6. This selection reduced in some cases drastically the number of viable pathways. For instance, for peri-urban SES, this subset of pathways requires transit as fast as possible toward a combination of CCA4|KCA9 when climate stress level is the lowest (Fig. 6A, option of arrangement CCA4|KCA9: see Table S1 in Supplemental S2 for details); or to transit through CCA4|KCA8 when climate stress level is the greatest (Fig. 6C, option D.1). Such drastic transformations are predicted to have large benefits, as they lead to a state where all the other governance pathway options become viable by 2040–2050. Interestingly, at climate stress level 1 (Fig. 6B), there is still a great diversity of highly secured viable pathways that sustain the required levels for all ES to be viable.

Figure 6 Most secured DAPP maps.

These maps represent the 10% most secured options of viable trajectories that respected the (normalized) sum of constraints for all ecosystem services. See Table S1 from Supplemental S2 for the exact definition of the nine CCA, KCA and associated OCA.

Similarly, in the rural SES, the most secured pathways lead a reduction in the transition time to CCA2|KCA3 for climate stress level 0 (Fig. 6D) or toward CCA3|KCA5 for climate stress level 2 (Fig. 6F). Subsequently, securing adaptation are achieved by further shortening the time required to transition to CCA4|KCA9 (Figs. 5D, 5F).

The model unexpectedly predicts that actors from the two SES will have a larger range of secured options at different time scales under climate stress level 1, as opposed to under milder or harsher conditions (Figs. 6B, 6E). This appears particularly true for peri-urban SES (Fig. 6E).

Switching pathways to prioritize different ES and resolve actor conflicts

Optimal DAPP maps (Fig. 7) were derived from Fig. 6, and showed viable adaptation pathways that maximize one ES+ at a time (or minimize one ES−). For example, in peri-urban SES with minimal level of climate stress (Fig. 7A), we predict that transiting directly to CCA4|KCA9 (as Fig. 6A would suggest to do to be more secured) will minimize the costs of maintenance but without maximizing the other ES+ (or minimizing the other ES−). The target of minimizing environmental hazards rather requires to delay the transition to CCA3|KCA7 (i.e., by creating a C-role) then the arrangement CCA2|KCA3 (i.e., contracting for PES). Maximizing all the other ES+ would require to transit first through the arrangement CCA3|KCA7, and then either CCA2|KCA3 or CCA4|KCA8.

Figure 7 Optimal DAPP maps.

These maps represent the subset of the most optimal sequence of institutional adaptation for each of the seven ecosystem services to be maximized or minimized. See Table S1 from Supplemental S2 for the exact definition of the nine CCA, KCA and associated OCA.

These results highlighted how the choices of pathways optimizing one ES+/ES− can impact other ES level through trade-off effects (Fig. 8). For instance, the pathway that maximizes pollination as priority objective (i.e., through pathway CCA1|KCA1(2020) → CCA3|KCA7(2020–2030) → CCA2|KCA3(2030–2040) → CCA4|KCA8(2040–2050) in Fig. 7A), is expected to produce positive, thus synergistic, effects by reducing environmental hazards (brown line) and maximizing fruit production (Fig. 8A). This pathway is also predicted to have negative impacts on wood biomass production (green line), sunlight protection (blue line) and landscape aesthetics (orange line). Consequently, new winners and losers relative to the ES are expected to emerge with changing pathways and thus according to the ES+ or ES− that are prioritized. Conversely, if actors seek to minimize the ES− “cost of maintenance” (Fig. 7F), then the most optimal adaptation pathway involves the following transition: CCA1|KCA1(2020) → CCA4|KCA9(2020–2040) → CCA1|KCA1(2040–2050) (Fig. 7A). This should result in a continual parallel decline of all the other ES+ and ES− over the next 30 years (Fig. 8F), alleviating the risks of ES trade-offs.

Figure 8 Predicted temporal dynamics of seven ecosystem services (ES+) and disservices (ES−) under the most optimal adaptation pathways, either maximizing ES+ or minimizing ES−, as indicated by the labels.

For example, (A) shows that the maximized ES+ is pollinator resources (thicker trend line), while the other ES+/ES− are allowed to vary (thinner trend lines). Results are based on the optimal DAPP map from Fig. 5A for a peri-urban SES under minimal climate stress. Supplemental S5 provides results for the other five DAPP maps from Figs. 7B–7F.

Sensitivity of DAPPs to changes in climate stress, SES and hedgerow types

Set of viability solutions according to the proportions of hedgerow types

We represented the viability kernel in the hedgerow state-space (Fig. 9) and examined how its size and shape—proxies for the number and types of adaptation pathways—were influenced by the proportion of different hedgerow types: species-poor hedgerows (PH) vs species-rich hedgerows (RH), and tall hedgerows (TH) vs short hedgerows (SH).

Figure 9 Influence of level of hedgerows species richness on the size and shape of the viability kernel.

2D cut of the four-dimensional viability kernel, expressed as a function of species-poor hedgerows (PH) and species-rich hedgerows (RH) for the two types of social-ecological systems (rural, peri-urban) and the three levels of climate stress. Blue dots close to the viability kernel limit mean that there are less options to adapt the hedgerow network through different nested governance arrangements changes, and thus are less secured.

We found that its shape and size were bound by minimal and maximal proportions of every hedgerow type, but that the proportions differed with climatic stress levels and SES types. For peri-urban SES, viable pathways were found under greater climatic stress, with a large range of proportions of RH, but a narrower one for PH (Fig. 9). By contrast in rural SES, greater climatic stress levels fit with viable pathways with much narrower ranges of proportion of both PH and RH. Patterns were somehow similar when considering species diversity to describe hedgerow types (Fig. S9 in Supplemental S5).

Overall, as climate stress increases, maintaining large diversity of viable pathway options should require relatively more RH than PH (Fig. 9) and more TH than SH (Fig. S9), regardless of the type of SES. If the state of the hedgerows remain constant (blue dots close to the threshold in Figs. 9 and S9), then greater climate change would put both SES types in an unsecured state, reducing drastically the number of possible adaptive actions, and increasing the risk of being trapped into a non-viable state (Figs. 9C, 9F and S9C, S9F).

Expected security gains per ES from switching hedgerow management targets

ES+ and ES− depended for a large part on the hedgerow species richness and height (see model in Table S2). We analyzed retrospectively whether switching actions from one hedgerow type to another may lead to more or less security gains, expressed as a distance to the limits of the ES satisfactory domain Kr (within ViabKR) for every ES+ and ES−, following methodological step 6. This was done for both the viable and non-viable adaptation pathways. We then analyzed whether the results contrasted when changing SES type and climatic stress level. The analysis revealed three consistent patterns of ES security shared between the two SES types (Figs. 10 and S10).

Figure 10 For every ecosystem service (ES), measure of the security gains ΔESTH-SH relative to the limits of the ES satisfactory domain (i.e., distance from the blue baseline), when switching the target of management from species-poor hedgerows.

As per the explanation in the method section 2.6.2, ΔESRH-PH > 0 indicates greater security gains when acting more on RH than on PH; whereas ΔESRH-PH < 0 indicates the opposite. The violins represent the probability density of ΔES values (with median in red) associated with all the adaptation pathways that are both viable and non-viable (yellow violins), or viable only (green violins). ES full name are provided in Fig. 8. Similar results are presented in Fig. S10 of Supplemental S5 between tall and short hedgerows.

Viable pathways (green violins) that increased relatively more the proportion of RH than PH (Fig. 10) were likely to build greater security in the viable ES+ provisioning (for pollination, fruit production, biomass production and sunlight protection). Conversely, viable pathways that increased relatively more the proportion of PH than RH (Fig. 10) were also likely to build greater security for keeping ES− (i.e., maintenance costs, environmental hazards) within the satisfactory threshold. In the absence of additional climate stress, we found only limited contrasts between viable and non-viable pathways (Figs. 10A, 10D and S10A, S10D). Increasing climate stress levels was predicted to amplify existing patterns of ES security, by pushing for even more RH for ES+ and even more PH for ES−.

Similar pattern of viability was observed for ES+ security when increasing TH over SH and for ES− when increasing SH over TH (see Fig. S10 in Supplemental S5). This pattern of priorities between hedgerow types was consistent across climate stress levels 0 and 1, but reversed at climate stress level 2 (see e.g., Fig. S10C for environmental hazards).

In the particular case of landscape aesthetics, ES security gains were not impacted by changing hedgerow types, as it was defined as the Shannon index of the diversity of all hedgerows, and was thus insensitive to changing any one type of hedgerow (see Table S2).

Discussion

DAPP maps enriches our predictions of potential adaptation options

Previous studies on the use of DAPP maps highlighted the importance of integrating detailed information on the SES context of adaptation (e.g., Stanton & Roelich, 2021). However, these studies also indicate that guidelines for organizing this information in a way that reflects the hierarchical nature of SES management.

Our goal was to demonstrate how to construct information-rich nested DAPP maps using Ostrom’s (1990) governance theory. We had two complementary objectives. The primary was practical: to design effective decision-support tools. The second was theoretical: to question whether and how the nature of abstract structures—such as nested adaptive institutional arrangements—could explain changes that manifest in the state, structure and dynamics of empirical phenomena, such as biodiversity, ecological infrastructures (hedgerows) and the delivery of ecosystem services to a diversity of actors.

To achieve this, we developed a method grounded in Ostrom’s main frameworks (i.e., IADF, SESF and CISF). Using these, we mathematically derived a social-ecologically rich set of adaptation actions and predictive, viable DAPP maps. We applied this method to analyze the long-term sustainable provision of ES produced by hedgerows in two SES, under climate change.

The results emphasized the influence of the SES type (rural vs per-urban) and climate stress levels (three drought levels) on possible pathways of nested governance adaptation supporting ES provisioning. We found that a wide diversity of possible nested arrangements and transitions between them were viable (among the ~5 million possible pathways), highlighting the great flexibility for diverse actors to meet their needs.

This nested DAPP approach demonstrates strong potential for tackling unique and complex SES adaptation challenges, while maintaining a sufficiently generic framework. We discuss the strengths and limitations of this method in the context of our case study.

Nested structure of DAPP maps enables more incremental adaptation options

Previous DAPP approaches, applied on complex SES involving biodiversity and ES, were defined at distinct levels of governance (e.g., Colloff et al., 2016; Lavorel et al., 2019, 2020; Bruley et al., 2021; Bergeret & Lavorel, 2022). The added-value of our integrated method was to show how to generate nested DAPP maps between these levels, involving adaptation actions between OCAs, within and between different KCAs, CCAs, for two SES characterized by different MCAs.

In the archetypical SES we presented, most actors who possessed hedgerows primarily used the nested governance structure CCA1|KCA1 (with a fixed OCA detailed in Supplemental S2). Our results indicate that this polycentric arrangement may not be sustainable in the long term, especially if climate change impacts hedgerows that are tall and biodiverse.

For rural actors, the most secured nested arrangement to stay in a viable pathway (Fig. 5) would be to transform CCA1|KCA1 into the most complex and polycentric CCA4|KCA8 (see Fig. 2 and further details in Table S1 from Supplemental S2). Even if we predict a permitted delay of few decades for this transition, in practice a direct CCA1 → 4 transformation should be very costly (for economic, technical, social or even cultural reasons). It could be anticipated from the required substantial disparities between CCA1 and CCA4 in collaboration skills, mutual trust, and the willingness to delegate power and roles among E, C, and P roles (Ostrom, 1990; Ban et al., 2015; Anderies, Barreteau & Brady, 2019). Changing KCA or even OCA would thus require less drastic investments for adaptation than changing CCA, and thus are expected to be adapted more regularly. This was confirmed by the semi-structured interview of actors in our study site (Supplemental S1). But we lacked sufficient data to evaluate the specific costs involved in the transition between different CCA and KCA.

Factoring in these costs may yield nested DAPP maps that better align to the need for more incremental transition pathways, similar to what we predicted in the peri-urban DAPP maps. For example, transitioning through CCA1 → 2 or CCA1 → 3 first, rather than directly from CCA1 → 4.

The method produced complex tipping points in adaptation pathways

The DAPP framework was initially designed with questions of systemic robustness in mind (Haasnoot et al., 2013; Haasnoot, Warren & Kwakkel, 2019), but not for a great number of nested actions and levels of systemic robustness and spillovers.

Here, we compared the effects of actions on emerging DAPP responses between two SES (peri-urban and rural), whose differences primarily rested on the constraints of ES levels for actors’ satisfaction and the impact of climate change on hedgerows between climate stress levels. All other factors were kept the same because the data we gathered and analyzed through the SESF did not provide enough evidence to detect significant differences between the two SES (see Supplemental S1).

Remarkably, by simply adjusting ES needs to the SES context and the climate stress impacts on hedgerows, we observed the emergence of entirely distinct DAPP maps. These maps had different patterns of transition pathways (Figs. 5–7) and of associated trade-offs and synergies among ES resource users (Fig. 8). The complexity of the patterns was akin to the systemic nature of the CISF representation, as expected by Anderies (2015). DAPP maps also emerged different patterns of viable (robust) pathways (Fig. 5). We also demonstrated how the adaptation choices involved changes in the relative proportion of species-rich/poor and short/tall hedgerows (Figs. 9, 10, S9, S10).

Some of these results were particularly unexpected. For instance, a slight shift in ES preferences between rural and peri-urban SES led to even more pronounced differences in diversity and priorities of pathway options and ES dynamics. Additionally, compared to the rural SES, the best options for securing the viability of peri-urban SES involved a swifter transition to the more intricate CCA4, with joint arrangements between private, community, and public sectors (see Table S1 for details). Another intriguing finding was the fact that moderate climate stress (level 1) diversified adaptation options, while severe stress (level 2) significantly limited them.

These emerging patterns of diversification or simplification of pathway options, along with others detailed in Supplementary Results S5, remain not fully understood, even though our analyses in Figs. 10 and S10 provided some insights into the key hedgerow types and ES influencing these changes. This highlights the need for further analyses on how changes in actions, hedgerows, and ES production affect pathway diversification.

Perspectives for using this approach for more complex SES

In our study, we considered seven ES as (in)tangible resource units, flowing from the R compartment to the exploitation role compartment E (through action set U1b in Fig. 1). Following the Millennium Ecosystem Assessment (MEA) (2005), these ES represent mostly provisioning ES (wood, fruits, etc.) and cultural ES (landscape aesthetics). However following the MEA, two other ES categories are worth considering, namely supporting and regulating ES. More recently, Colloff et al. (2016) developed the concept of ‘adaptation services,’ which refers to specific types of supporting or regulating services that enhance the resilience of social systems by providing climate-related benefits, such as protection, resilience, or enabling adaptive options.

The CISF is valuable because it streamlines the modeling of ES from interconnected ecological infrastructures, while also highlighting how the chain of ES and infrastructures can resemble collective-choice arrangements (KCA), a concept emphasized by researchers like La Notte et al. (2017) and Lavorel et al. (2019). For example hedgerows, as semi-natural infrastructures provide supporting ES like pollination and pest control (akin to action set U4a), which enhance crop growth (U0a), and regulating ES such as water flow management (U5a). If biodiversity in hedgerows or grasslands adjusts to climate change, their effects could impact the C, R, and E states, similar to a KCA. The same approach can be used to enrich DAPP studies, like those presented by Lavorel et al. (2019). For instance in their article, grasslands can be described as having a C-role through the control of erosion on adjacent fields (i.e. using action set U5a). Similarly carbon storage, fodder resilience, and services like aesthetics and shade can be defined using different action sets (U0a, U4a, U1b, U6b). Lastly, services like connectivity and transformability are connected to the R and C-role compartments, and transformability can be modeled as the acceptable range of structural, compositional or entropic2 changes in R or C tolerated by other roles.

Together these examples suggest that the CISF (and thus by association the SESF through Fig. 1) could flexibly be used to model potentially very complex KCA, and by extension DAPP problems, involving complex networks of roles attributed to ecological infrastructures, species, actors and ES (cf. Vogt et al., 2015; Partelow & Winkler, 2016; Rova & Pranovi, 2017).

Conclusion and perspectives

Our objective was to develop DAPP maps enriched with socio-ecological information, integrating Ostrom’s frameworks (IADF, SESF, and CISF) with concepts from complex dynamical systems and viable control theories. This approach enabled the creation of innovative maps that balance three key dimensions of SES adaptation: (i) identifying socio-ecological attributes via SESF, (ii) modeling systemic risks and robustness through CISF and viability theory, and (iii) mapping nested governance-adaptations through IADF and CISF.

By applying this framework to one case study, we demonstrated how viable short- and long-term adaptation pathways and ecosystem service (ES) outcomes emerge under the influence of governance pathways, SES types, and climate stress levels. Our findings underscore the importance of defining SES targets, actor roles, infrastructure vulnerabilities, and nested governance structures in predicting these patterns.

A key contribution of our approach lies in achieving logical completeness between Ostrom’s theoretical frameworks, viability theory and DAPP maps, providing a deeper understanding of the socio-ecological elements that constrain or foster adaptiveness. However, despite its theoretical strengths, practical challenges remain for actors seeking to operationalize this approach in governance planning under climate change. Questions about its usability and alignment with actors’ life-history constraints (cf. Stanton & Roelich, 2021) merit further exploration, especially given the growing adoption of DAPP maps by policymakers.

Future work should focus on refining and testing these novel maps to ensure they are both practical and widely accessible. This includes enabling scientists and local actors to apply them effectively for SES adaptation or to test elements of Ostrom’s governance theory in adaptive contexts. As a next step, we propose building on insights from the integration of serious gaming methods with DAPP maps (Blackett et al., 2022) to further test the incorporation of elements developed in our article.

Supplemental Information

Supplemental Information 1 Raw social-ecological data and 4th tier SES analysis.

This compressed file contains: (1) A spreadsheet Presenting the details of the 4th tier SESF analysis, (2) A pdf file of the semi-structured interview guide that was used to obtain some of the SES variables. The unstructured part of the interviews contained identifiable personal information and thus was removed. It can be shared on demand only for actors who sign a sharing-consent agreement. (3) A pdf file in French language of the original M.S. report presenting elements of social-ecological analysis of the study site that was conducted during 2021. It includes the analysis of local and regional gray literature and anonymized interviews with some local stakeholders. This data represent one of the source of information for performing the socio-ecological system (SES) framework analysis across the two sites under investigation. The analysis of stakeholders encompassed both qualitative and quantitative data, facilitating an examination of their capacity to identify various types of hedgerows, their relationships with these features, preferences regarding ecosystem services associated with hedgerows, and their inclinations toward different governance arrangements and developmental trajectories. Details regarding the survey methodology and utilized documents are provided within the document. The Author of the report is Elise Krief, M.S.

Supplemental Information 2 Details of data, GIS analysis and survey documents used for constructing the modeling the structure and dynamics of the hedgerow network in the study site.

Details of the methodological implementation for the SES case studies in the French Auvergne Region. It also includes Tables S1 and S2.

Supplemental Information 3 Raw physical and ecological data, GIS analysis and survey used for modeling the structure and dynamics of the hedgerow network in the two sites.

S3.a. The orthophotographies with the distribution of hedgerows (orange color) for the two sites (rural and peri-urban) and four years (1946, 1989, 2009, 2019). QGIS was used to generate these pictures. S3.b Spreadsheet Summary of GIS Information for Hedgerows Across the peri-urban study site (city of Veyre Monton). The spreadsheet provides a comprehensive summary for each hedgerow located within the study site of (1) Geographic Information System (GIS) attributes, (2) the physical attributes,and (3) the ecological attributes from the survey conducted in summer 2021. The GIS attributes are GPS coordinates, identification (ID), and hedgerow typology for the years 1989, 2000, and 2019. The physical attributes of every hedgerow encompass various features including size, height, length, adjacent habitats, presence of embankments, stone walls, ditches, and number of plant layers. Ecological attributes encompass a range of factors such as the number of plant species, plant pollinating species, fruit species, living wood biomass, snag number, and deadwood biomass. These metrics offer crucial information about the biodiversity and some ecosystem services characterizing the hedgerows. The data compiled in this spreadsheet serves as a valuable resource for analyzing the ecological dynamics and conservation status of hedgerows in the study area, facilitating informed decision-making and management strategies for their preservation and enhancement. This information also served as the foundation for analyzing the annual probability of transition between various combinations of hedgerow types, which include hedgerows that were tall species-rich, short species-rich, tall species-poor, short species-poor, and none. It is noteworthy that the hedgerow data from the 1946 aerial photography was exclusively utilized to estimate hedgerow density. Due to the limited visual quality, determining the exact hedgerow type was not feasible. However, this data significantly contributes to understanding hedgerow dynamics and transitions over time. S3.c. a link to the necessary QGIS data (1.6Gbytes of data) to perform the QGIS analysis of the hedgerow network for the two sites (rural = La Sauvetat, peri-urban = Veyre-Monton) for years 1946, 1989, 2000, 2019, nearby Clermont-Ferrand metropolis. This compress file contains the aerial orthophotographs (tiff file, resolution: 0.2 m per pixel), the QGIS layers of the land-use, of the hedgerow network, and the typology of hedgerows. Aerial orthophoto for year 2000 could not be shared and can be downloaded by following the link to the original data source: https://ids.craig.fr/datahub/dataset/0008b161-9db6-4694-8b4e-3e0ea4fdb934.

Supplemental Information 4 Octave/Matlab code that can be used to reproduce the results of this study.

Supplemental Information 5 Results on tall and short hedgerows (Figures S9 and S10) associated with Figures 9 and 10.

Archetypal results for the nine pathways where the CCA and KCA have been fixed and do not change during 80 years.

We thank Johannes STEIGER for early discussions and advices on actors and local SES. We express our gratitude to Pr. J. Marty Anderies for engaging in fruitful discussions regarding the use of the CISF, ensuring its alignment with our present objectives. We thank Dr. Magali Weissgerber, and master students Elise Krief, Lisa Minaca, Quentin Guillois and Loïc Montel for their contribution to, respectively, interviews with local actors, field hedgerows surveys, aerial photographs analyses and GIS maps elaboration, as well as SESF analyses.

Additional Information and Declarations

Competing Interests

The authors declare that they have no competing interests.

Author Contributions

Jean-Baptiste Pichancourt conceived and designed the experiments, performed the experiments, prepared figures and/or tables, authored or reviewed drafts of the article, and approved the final draft.

Antoine Brias performed the experiments, analyzed the data, prepared figures and/or tables, wrote the code of the model, and approved the final draft.

Anne Bonis conceived and designed the experiments, performed the experiments, authored or reviewed drafts of the article, and approved the final draft.

Data Availability

The following information was supplied regarding data availability:

The raw data are available in the Supplemental Files and at Anonymous Authors. 2024. Integrating Adaptation Pathways and Ostrom’s Framework for Sustainable Governance of Social-Ecological Systems in a Changing World DOI 10.57745/IVWYL4.

1 See Martin, Deffuant & Calabrese (2011) for the resilient case.

2 cf. Greek term “tropos” signifying transformation.

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
