# Peer review of "Integrating adaptation pathways and Ostrom’s framework for sustainable governance of social-ecological systems in a changing world"

_PeerJ, doi:10.7717/peerj.18938_

## Round 0.1 · original submission · Major Revisions

Two expert reviewers have evaluated your manuscript and their comments can be seen below and in an attached PDF. As you will see both comment on the length and the complexity of the dcoument and have made a number of suggestions that should help you to improve the presentation and content of the manuscript. Please ensure that you follow their suggestions and prepare a revised manuscript. Also, provide a rebuttal letter that clearly identifies what modifications have been made and where they can be found.

Reviewer 1 ·

Basic reporting

Generally, the paper is very comprehensive in my view employing multiple theories and concepts (SES, CIS and nested governance) to enhance the existing DAPP framework for potential pathway simulations of hedgerows. I think the paper is rather long (about 50 pages!) and quite hard to follow probably due to the current writing structure. While i try appreciate the emphasis and contribution of the paper, the current structure or writing style needs to be improved.

Firstly, is current DAPP framework insufficient? not adequate to tackle current adaptation issues? clearer and strong rationales are required to justify the study's need particularly highlighting the real and tangible issues of the current framework which have propelled the authors to use various models and theories to hopefully provide pragmatic solutions to policymakers, not just remain theoretical. Below are some specific concerns that need to be addressed.

On nested governance, Ostrom's final design principle.. The idea of polycentricity or devolution needs to be embedded.

-see line 138-140, why they are being aligned to different timescale? on what basis that OCA is short term, KCA is mid term and CCA is Long term?

Minor Proofreading is necessary due to typo and spelling errors. Some texts have been written colloquially (see section 4.3.1). Kindly revise

Experimental design

I urge authors to have a methodological framework on how DAPP since it involves several steps with many equations. what software was used for the pathways simulation?

Validity of the findings

Though the validity has been obtained based on the theoretical and mathematical models, i suggest that some forms of interview surveys can be conducted to verify and validate the generated results (especially the practicality and accuracy as to whether or not they are reflective of the current and potential/future scenarios)

Additional comments

No additional comments

Reviewer 2 ·

Basic reporting

The paper is long and complex with lots of ideas - combining multiple frameworks, mathematical models and viability theory. This is a lot to digest. The main messages of the paper need to be made more clearly and earlier. Some of the material might best be moved to a supplementary materials section.

Also, please put the figures in line for review purposes or send a "review version" of the manuscript as it is very difficult to review with the figures at the end.

Experimental design

Not applicable - the study does not involve an experiment.

Validity of the findings

The article suggests the need for "renewal" or "extensions" of the SES and CIS Frameworks. Perhaps this is a matter of word choice, but the text reflects some lack of clarity regarding what is meant by "framework", "theory" and "model" in the Ostrom tradition. Given that the paper leverage's Ostrom's frameworks (the SES and CIS are deeply related and are both derivatives of the IAD) this needs to be clarified. This raises two points:

1) Much of what are described as "extensions" or "renewals" of these frameworks are not. They are simply the result of APPLYING the frameworks - i.e. they are descriptions of models elicited by the frameworks. Frameworks are very general and describe only the essential and most basic features of a system. The break down of different PI actors in the paper is an example of a MODEL of how the PI functions in a particular context. I think it is critical that the manuscript be revised to appropriately reflect that the authors are using the SES and CIS to explore key elements of DAPPs.

2) As mentioned above, the SES and CIS are both derivative frameworks of the IAD. In fact, the temporally linked action situations (a sequence of adaptive policy decisions is a set of linked action situations) in the DAPP map very naturally on to the IAD. Then specific features of the SES and CIS can be leveraged appropriately in relation to unpacking aspects of the IAD. Making this link is critical - as well as noting the rule clusters that accompany the IAD. This will avoid confusion about, for example, agents with multiple roles as not being captured in the frameworks. The RU, PIP, and PI elements do not define agents. They define roles which, in turn, are defined by institutions and norms (from the rule clusters). In almost all cases, agents play multiple roles.

I have attached an annotated PDF with further comments along these lines.

Annotated reviews are not available for download in order to protect the identity of reviewers who chose to remain anonymous.

---

## Round 0.2 · accepted · Accept

I am satisfied with all of the changes that have been made to the manuscript and am recommending that it be accepted for publication in PeerJ. Congratulations.

Reviewer 2 ·

Basic reporting

No comment

Experimental design

No comment

Validity of the findings

No comment

Additional comments

The authors have addressed my concerns in their revision.